# Transforming Radiology Workflows: Pretraining for Automated Chest X-ray Report Generation

**Shashank Gupta**\*                                          SHASHANK.GUPTA@UKY.EDU

**Yuhang Jiang**\*                                              YUHANG.JIANG@UKY.EDU

**Abdullah-Al-Zubaer Imran**                                   AIMRAN@UKY.EDU

*University of Kentucky, Lexington, KY, USA*

## Abstract

Automated chest X-ray report generation using machine learning has emerged as a promising technology for improving the accuracy and efficiency of chest X-ray interpretation. In this paper, we present a novel approach for automated report generation that combines the power of vision transformers for image information encoding and PubMedBERT for text decoding. Our model extracts image features using a vision transformer and text features using PubMedBERT. The encoded features are then fed into a text decoder to generate standardized reports. We trained our model on a dataset of chest X-rays and corresponding report findings (a subset of the MIMIC-CXR dataset) and evaluated its performance on a small subset of the IU dataset.

**Keywords:** Chest X-ray, BLIP, PubMedBERT, ViT, Pre-Training.

## 1. Introduction

Chest X-rays are widely used for diagnosing chest-related conditions but require specialized expertise for interpretation, which can be time-consuming and subject to errors. Manually writing every report is also costly, prone to variability, and may delay treatment. Healthcare professionals may interpret the same image differently, leading to inconsistent diagnoses and delays in treatment. Recent advancements in machine learning may improve the efficiency and accuracy of chest X-ray interpretation by automating the report-generating process which can be helpful for reducing the workload of radiologists and facilitating quick diagnoses. This could reduce wait times for patients, minimize errors, and make interpretation more accessible while also being cost-effective.

In this research paper, we present a novel machine-learning model for generating chest X-ray reports. Our model utilizes a vision transformer (ViT) (Dosovitskiy et al., 2021) to extract features from the chest X-ray images, followed by a text decoder to generate standardized reports. The reports include key features of the X-ray image, such as lung function, the presence of any abnormalities, and a differential diagnosis based on the identified features. We train our model on a subset of the MIMIC-CXR dataset of chest X-rays and corresponding reports and evaluated it on the IU dataset.

Similar work has been performed by researchers in the past. (Wu et al., 2022) presents DeltaNet for automatically generating medical reports which applies a conditional generation process. (Najdenkoska et al., 2021) proposes variational topic inference for automatic report generation by introducing a set of topics as latent variables to guide sentence generation by aligning image and language modalities in a latent space. (Liu et al., 2021) proposes a Contrastive Attention (CA) model for X-ray report generation.

---

\* Contributed equally

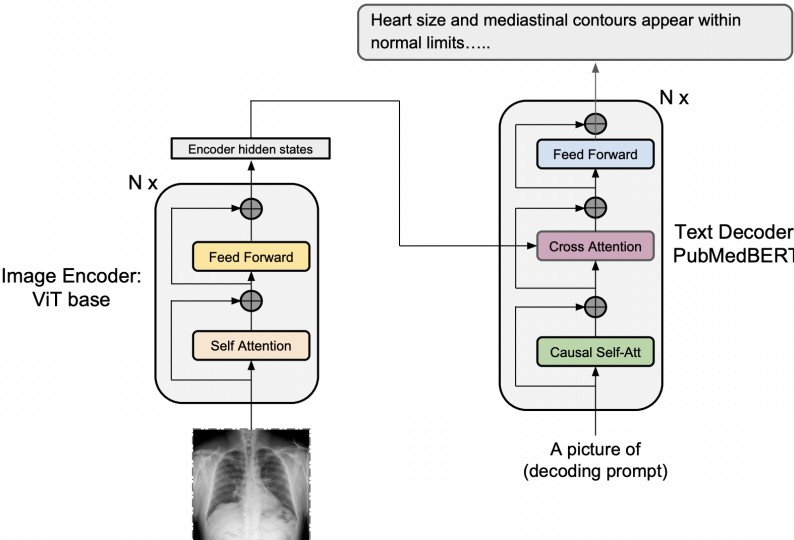

Figure 1: The encoder-decoder architecture of our image-generation framework

## 2. Methodology

We adopt the architecture from BLIP (Li et al., 2022), a bootstrapping language-image model pre-trained for both understanding-based and generation-based objectives. The model pre-training takes the input of pairs of images and the corresponding text and afterward generates radiology reports for given X-ray images. The framework uses a multimodal mixture of encoder-decoder (MED) model architecture that enables effective multi-task pre-training. MED can operate as a unimodal encoder, an image-grounded text encoder, or an image-grounded text decoder. Our model is jointly pre-trained with three vision-language losses: image-text contrastive learning, image-text matching, and image-conditioned language modeling. For image-text contrastive learning loss and image-text matching loss, we follow by (Li et al., 2021). We use the language modeling loss which is a cross-entropy loss for training the model to maximize the likelihood of the next token in the text.

### 2.1. Losses

In this work, we pre-train a BLIP model from the beginning with X-ray images and corresponding findings from reports. Our model architecture employs a visual transformer (ViT) as its image encoder, which divides an input image into patches and encodes them as a sequence of embeddings. Additionally, a [CLS] token is used to represent the global image feature. The text encoder is initialized with PubMedBERT(Gu et al., 2020) which is pre-trained on PubMed abstracts. To serve as a framework for text generation, our model replaces the bi-directional self-attention layers in PubMedBERT with causal self-attention layers that can operate as an image-grounded text decoder. At inference time, our model provides an encoder-decoder architecture for generating radiology reports with a given X-ray image, which is shown in Figure 1.

Table 1: Scores calculated on IU dataset.

| Method | Model | Jaccard | ROUGE-2 | ROUGLE-l | METEOR |
|---|---|---|---|---|---|
| Fine Tuned | DeltaNet-3C(Wu et al., 2022) | - | - | 0.379 | - |
| | TieNet (Wang et al., 2018) | - | - | 0.226 | - |
| | CvT-212DistilGPT2(Nicolson et al., 2022) | - | - | **0.376** | 0.200 |
| Zero Shot | ViT + PubMedBERT (ours) | 0.14 | 0.07 | 0.24 | **0.23** |

## 3. Experiments and Results

We utilize the MIMIC-CXR dataset (Mechanical Ventilation, Vital Signs, and Clinical Data Chest X-Ray)(Johnson et al., 2019), which is a large, publicly available dataset of chest X-ray images with corresponding radiology reports to pre-train our model. We sampled from the original dataset, and the resulting dataset consists of 12,676 images with corresponding reports. To create captions for our images, we use the findings from the report, rather than the impression, as they provide a more objective description. We select frontal X-ray images for pre-training as they contain more informative features. Our model was trained using image-text contrastive loss, image-text matching loss, and language modeling loss, with the same objective as BLIP to improve language generation. We use ViT-base as our image encoder and PubMedBERT-base as our text decoder. We resize all the images to 224 X 224 and pre-train our model for 100 epochs with batch size and initial learning rate of 3e-4 with 3000 warm-up steps.

To evaluate the performance of our pre-trained model, we selected the IU dataset (OpenI) which contains 3,307 frontal images and corresponding findings which can be obtained from Kaggle. We then compared the system-generated findings with the original findings in the reports and calculated various metrics, including Jaccard similarity, ROUGE, and METEOR scores to measure the accuracy and quality of the generated reports. The scores are displayed in Table 1 compared to other fiine tuned methods.

## 4. Conclusions

Our pre-trained model is aimed at generating X-ray reports, which can be helpful for reducing the workload of radiologists and facilitating quick diagnoses. To this end, we employed the BLIP architecture, which is known for its high accuracy and efficiency. The image encoder we used is a Vision Transformer, which has shown promising results in computer vision tasks, while the language encoder we used is PubMedBERT, a pre-trained language model specifically designed for biomedical applications.

While our current pre-trained model has shown some promise, its performance is limited due to the small size of the pre-training dataset. However, we believe that using the full MIMIC-CXR dataset for pre-training will greatly improve our model's performance and accuracy.

By utilizing the full MIMIC-CXR dataset, which will provide us with a much larger and more diverse set of training data, we hope to achieve higher accuracy and more robustness in our model, which will make it a more useful tool for radiologists and medical professionals.

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
