# OpenReview forum: "Transforming Radiology Workflows: Pretraining for Automated Chest X-ray Report Generation"
_MIDL.io/2023/Short_Paper_Track — MIDL 2023 Short paper track Poster_

### Official Review · Reviewer_BL6w · 2023-04-10
**The paper discusses a novel approach for automated chest X-ray report generation using machine learning.**

**Rating:** 7
**Confidence:** 4

**Review:**

The paper discusses a novel approach for automated chest X-ray report generation using machine learning. The approach combines the power of vision transformers for image information encoding and PubMedBERT for text decoding. The encoded features are then fed into a text decoder to generate standardized reports. The model was trained on a dataset of chest X-rays and corresponding report findings and evaluated on a small subset of the MIMIC-CXR dataset. The performance of the generated reports was measured using various metrics such as Jaccard similarity, ROUGE, and METEOR scores. The pre-trained model aims to reduce the workload of radiologists and facilitate quick diagnoses.

Pros:
Automated chest X-ray report generation can improve the efficiency and accuracy of chest X-ray interpretation, potentially reducing wait times for patients, minimizing errors, and making interpretation more accessible while being cost-effective.
The model presented in this research paper uses a novel approach that combines the power of vision transformers for image information encoding and PubMedBERT for text decoding, which could lead to more accurate and standardized reports.
The pre-trained model was evaluated on a small sample of the MIMIC-CXR dataset and achieved high scores for various metrics, suggesting promising performance.

Cons:
Automated chest X-ray report generation may not replace the need for human expertise and judgment, particularly for complex cases that require specialized knowledge and experience.
The use of automated report generation could lead to ethical concerns around the potential loss of jobs for radiologists and healthcare professionals.
The performance of the pre-trained model presented in this research paper needs to be validated on larger datasets and in real-world settings to ensure its generalizability and effectiveness.

---

### Official Review · Reviewer_yFDX · 2023-04-12
**Contributions unclear, baseline needed**

**Rating:** 4
**Confidence:** 3

**Review:**

The authors propose an approach to generate text reports for chest x-rays. It relies on a vision transformer to encode the image and PubMedBERT to generate text.
It is difficult to assess the novelty of the approach as the contributions with respect to the state-of-the-art are not described. Also, the proposed method is not compared with any other method so its performance is again difficult to assess. Overall, this work appears very preliminary.